# Prognostic, Diagnostic and Predictive Biomarkers in the Barrett’s Oesophagus-Adenocarcinoma Disease Sequence

**DOI:** 10.3390/cancers14143427

**Published:** 2022-07-14

**Authors:** Darragh O’Dowd, Jacintha O’Sullivan, Simone Marcone

**Affiliations:** Department of Surgery, Trinity Translational Medicine Institute, Trinity St. James’s Cancer Institute, Trinity College Dublin, St. James’s Hospital, D08 W9RT Dublin, Ireland; odowdd2@tcd.ie (D.O.); osullij4@tcd.ie (J.O.)

**Keywords:** Barrett’s oesophagus, oesophageal adenocarcinoma, biomarkers, screening, neoadjuvant treatment

## Abstract

**Simple Summary:**

Oesophageal adenocarcinoma (OAC) is a type of cancer of the oesophagus (food pipe) which is associated with poor patient outcomes. Barrett’s oesophagus (BO) is a precancerous condition of the oesophagus associated with chronic heartburn. Currently, surveillance programs exist which monitor patients with BO to prevent it from developing into OAC. However, these surveillance programs are expensive and unpleasant for patients. Prognostic biomarkers are signs which could be measured to determine the chance of someone with BO developing OAC, allowing more targeted surveillance. Similarly, diagnostic biomarkers are indicators which could be measured to see if someone has OAC. Developing new diagnostic biomarkers could allow wider population testing. Only a small proportion of patients with OAC respond to treatment before surgery. Predictive biomarkers could be measured to predict whether someone would respond to the treatments, allowing more individualized therapy. This review focuses on potential biomarkers which could improve patient outcomes in BO/OAC.

**Abstract:**

Oesophageal adenocarcinoma (OAC) incidence has increased dramatically in the developed world, yet outcomes remain poor. Extensive endoscopic surveillance programs among patients with Barrett’s oesophagus (BO), the precursor lesion to OAC, have aimed to both prevent the development of OAC via radiofrequency ablation (RFA) and allow earlier detection of disease. However, given the low annual progression rate and the costs of endoscopy/RFA, improvement is needed. Prognostic biomarkers to stratify BO patients based on their likelihood to progress would enable a more targeted approach to surveillance and RFA of high-risk precursor lesions, improving the cost–risk–benefit ratio. Similarly, diagnostic biomarkers for OAC could enable earlier diagnosis of disease by allowing broader population screening. Current standard treatment for locally advanced OAC includes neoadjuvant chemotherapy (+/− radiotherapy) despite only a minority of patients benefiting from neoadjuvant treatment. Accordingly, biomarkers predictive of response to neoadjuvant therapy could improve patient outcomes by reducing time to surgery and unnecessary toxicity for the patients who would have received no benefit from the therapy. In this mini-review, we will discuss the emerging biomarkers which promise to dramatically improve patient outcomes along the BO-OAC disease sequence.

## 1. Introduction

Barrett’s oesophagus (BO) is considered a precursor to oesophageal adenocarcinoma (OAC) and is defined as columnar metaplasia of the stratified squamous epithelium of the distal oesophagus which arises in the setting of gastro-oesophageal-reflux disease (GORD) [1]. Given the well-characterized link between BO and OAC, endoscopic screening for BO in high-risk individuals followed by regular surveillance of BO patients is recommended by many international medical societies [2].

Currently, screening for OAC occurs only as part of surveillance in those diagnosed with BO, while BO diagnosis relies on endoscopy on a background of GORD symptoms in high-risk patients [3]. Individuals at high risk for BO/OAC include those with GORD symptoms and multiple risk factors for disease, e.g., age > 50, male, raised BMI, and family history of BO/OAC [4]. Surveillance involves upper endoscopy with biopsy looking for BO, BO with dysplasia or OAC. There are important limitations in endoscopic surveillance as it is relying on invasive and costly techniques and on complex histopathological grading to make clinical decisions. Therefore, with the exception of patients presenting with high grade dysplasia (HGD) it remains difficult to identify which patients are likely to progress from Barrett’s to cancer [5]. The efficacy of radiofrequency ablation (RFA) in BO in preventing progression to OAC has been shown in randomized controlled trials [6]. The success of RFA in preventing progression of dysplastic BO has increased interest in stratifying non-dysplastic Barrett’s oesophagus (NDBO) to those with “high risk” disease that may benefit from RFA as this procedure carries some risks and has high costs for the healthcare system [7]. Currently, gastrointestinal (GI) society guidelines recommend endoscopic surveillance over endoscopic ablative therapy in the non-dysplastic population of BO [2].

Therefore, given the economic and human cost of oesophageal endoscopy/RFA and the fact that the annual risk of OAC development for a patient with BO is as low as 0.12–0.23% [8,9], prognostic biomarkers to stratify patients based on the risk of BO to OAC progression are needed to improve care. In fact, the development of a biomarker platform to discriminate BO patients at low risk of progression could represent a cost-effective way to manage patients with longer intervals in endoscopic surveillance or discontinuation of surveillance entirely. In addition, if prognostic biomarkers could clearly indicate that a BO patient is at high risk for cancer progression, then the balance between risks, costs, and benefits would be tipped in favor of endoscopic ablation for these specific patients.

The incidence of OAC in the Western world has rapidly increased over the previous three decades, surpassing oesophageal squamous cell carcinoma (OSCC) as the most prevalent histologic subtype of oesophageal malignancy in the developed world [10]. Despite this increase, the prognosis has remained relatively poor, with a mean 3-year survival rate across seven high-income countries of just 25.3% [11]. Reasons for this are multi-faceted but center on late stage at diagnosis (only approximately 25% of OAC present with localized disease [12]) and lack of effective treatments [12]. As more than 40% of those who develop OAC do not experience typical symptoms associated with GORD and BO [13], the discovery and validation of accessible diagnostic biomarkers would aid in earlier diagnosis, improving outcomes for OAC patients and could even alleviate the need for endoscopic surveillance.

Predictive biomarkers for response to treatment could prove similarly beneficial. Currently, the mainstay therapy for locally advanced disease includes neoadjuvant therapy (NA-T) followed by oesophagectomy [13]. However, only approximately 30% of patients achieve a pathological complete response to NA-T [14]. Thus, predictive biomarkers to stratify non-responders could improve the time to surgery and reduce unnecessary toxicity of NA-T in this cohort, thereby potentially improving survival.

In this review article, we have focused on recent studies which emphasize translational biomarker development specifically along the BO-OAC disease sequence, attempting to capture the variety of techniques being utilized in the field. This article provides a contemporary review of research in the development of biomarkers to improve surveillance, diagnosis and management of BO and OAC patients, identifying promising candidates and offering an insight into the current state of development of biomarkers which could improve outcomes in this cohort as illustrated in Figure 1.

## 2. Prognostic Biomarkers of Progression from Barrett’s Oesophagus to Oesophageal Adenocarcinoma

Patients diagnosed with BO undergo regular endoscopic surveillance despite the low annual conversion rate to OAC. Aside from the economic cost, patients also experience considerable discomfort and anxiety in relation to endoscopy [15]. Prognostic biomarkers could play a role in stratifying those with BO into risk categories to improve cost-effectiveness of surveillance, allow selection for RFA and alleviate unnecessary patient burden. Currently, the sole method of risk stratification in BO patients is by the degree of dysplasia [ NDBO, indeterminate dysplasia, low grade dysplasia (LGD), HGD, or carcinoma] detected on endoscopy [16]. The degree of dysplasia is used to dictate surveillance intervals and endoscopic interventions [16]. This histologic approach is subject to the limitations of sampling error and interobserver variability in dysplasia classification among pathologists [16]. Tissue molecular biomarkers are being examined as a means of overcoming histopathologist limitations. P53 immunostaining is recommended as an adjunct in the diagnosis of dysplasia by the British guidelines as a means of reducing interobserver variability although the American Gastroenterological Association maintains that the role of p53 in this context requires further clarification [3,17]. In a 2018 study, the addition of a corresponding p53 immunohistochemistry (IHC) staining to 60 examined H&E stained BO biopsy slides as examined by 10 GI pathologists was found to significantly increase both interobserver agreement and diagnostic accuracy (as compared with “gold standard” diagnosis via consensus agreement between five expert BO pathologists) (*p* = 0.071 and *p* = 0.0021) [18]. The results of this study are limited by the use of only one slide per biopsy “case”.

Aberrant p53 expression alone has also been assessed as a prognostic biomarker for risk of progression from BO/LGD to HGD/OAC. Kastelein et al. analyzed p53 expression (classified as normal, overexpressed or loss of expression) of formalin-fixed paraffin-embedded biopsies (FFPE) from 635 BO patients consisting of *n* = 586 non-progressors (those who did not develop HGD/OAC from baseline BO/LGD) and *n* = 49 progressors (those who went on to develop HGD/OAC from baseline BO/LGD) via IHC with consensus between two pathologists required [19]. Patient surveillance period had a median of 6.6 years and consisted of a median of four follow-up endoscopies. Aberrant p53 expression in NDBO/LGD was associated with an increased risk of progression with loss of p53 being more strongly associated with progression (risk ratio = 14) compared to p53 overexpression (risk ratio = 5.6). The prognostic sensitivity and specificity of aberrant expression was 49% and 86%, respectively. When aberrant p53 expression coincided with LGD, the risk of progression was even greater. Davelaar et al. added fluorescent in situ hybridization (FISH) analysis of p53 to IHC to improve the ability to detect p53 abnormalities and thus prognostic ability in a cohort of 91 patients, including *n* = 80 non-progressors (BO patients diagnosed with at most LGD following minimum 3 years surveillance) and *n* = 11 progressors (BO/LGD to HGD/OAC) [20]. Abnormal p53 expression, as detected by IHC and FISH, was found to be an independent predictor of progression. There were low levels of overlap of p53 abnormalities detected by the two methods, as highlighted by p53 abnormalities being detected in progressors as follows: five by IHC alone, two by FISH alone, two by both methods and two by neither method. Combining the methods led to a prognostic biomarker of progression with a sensitivity and specificity of 81.8% and 85%, respectively. While promising, the small cohort assessed limits the strength of these results. In a retrospective study including 24 progressors (developed HGD/OAC from baseline BO/LGD after >1 year of sampling) and 73 non-progressor BO/LGD patients (with >2 years follow-up since sampling) Stachler et al. used next-generation sequencing of one tissue sample from each participant to identify genetic abnormalities predictive of progression [21]. P53 genetic abnormalities were once again shown to be associated with increased risk of progression with mutations found in 46% of progressors but only 5% of non-progressors. While suggestive, the low number of progressors and use of only one tissue sample from each participant represent limits of this study.

In a nested case control study, Duits et al. looked at two biomarkers obtained from FFPEs and analyzed their predictive ability when combined with the presence or absence of LGD to distinguish progressors (those that develop HGD/OAC from BO/LGD at baseline) from non-progressors (patients diagnosed with at most LGD following minimum 2 years surveillance) [22]. The biomarkers measured were p53 (IHC) and aspergillus oryzae lectin (AOL) (IHC). Presence or absence of LGD was assessed by consensus diagnosis. All participants required a minimum of 2-year endoscopic surveillance before study end. Similarly, any patient with >T1 disease at diagnosis was excluded to ensure progressors were developing new disease. Of note, an expert consensus diagnosis of LGD was associated with a 34-fold increased risk of progression. On multivariate analysis, the panel demonstrated an area under the receiver operating characteristic (AUROC) of 0.73 and a sensitivity and specificity range of 41.1–88.9% and 88.1–35.8%, respectively, depending on the cut-off value selected. Aneuploidy was also intended to be accessed as a biomarker; however, the assay used was found to produce inconsistent results. One limitation of the panel is the inclusion of LGD, given the stringent and often unavailable resources (two expert GI pathologists) required for diagnosis and the fact that endoscopic ablation may be recommended with repeat findings [4].

Timmer et al. evaluated six molecular markers (p16, p53, Her-2/neu, 20q, MYC, and aneusomy) by DNA FISH on brush cytology specimens in a prospective cohort of 428 patients with NDBO [23]. Participants were followed up by endoscopic surveillance for a median of 43 months with a primary study outcome of disease progression (development of HGD or OAC). In total, 22 progressors were detected within the study with 13 developing OAC. By combining the three significant molecular markers of progression on univariate analysis (MYC gain, p16 loss and aneusomy) with age and circumferential BO segment length, a model to divide patients into high- and low-risk groups was developed. The model demonstrated a sensitivity of 0.86 and a specificity of 0.54 while the positive and negative predictive values were 9% and 99%, respectively. Using this model, 19 of the 22 progressors and 188 of the non-progressors would have been classified as high risk while 3 progressors and 218 non-progressors would have been classified as low risk.

Given that inflammation has long been implicated as a driver in the BO-OAC disease sequence [24], O’Farrell et al. investigated markers of oxidative stress as potential biomarkers to indicate likely progressors among BO patients [25]. The group used IHC to measure the expression of various proteins in tissue samples collected from *n* = 23 NDBO patients, consisting of 10 non-progressors (showed no evidence of disease progression over a median 7.5-year follow-up period) and *n* = 13 progressors (developed HGD/OAC over a median follow-up period of 2 years). The intensity and positivity of stromal and epithelial expression of 8-oxo-dG were found to be significantly weaker among progressors compared to non-progressors, counterintuitively suggesting that progressors have an environment of decreased oxidative stress. The authors propose that the explanation for this may be linked to mitochondrial dysfunction. Regardless, this study, though limited by the small sample sizes, suggests that 8-oxo-dG has the potential to identify those most at risk of malignant transformation. The potential prognostic biomarkers discussed are summarized in Table 1 below.

## 3. Diagnostic Biomarkers of Barret’s Oesophagus and Oesophageal Adenocarcinoma

A promising approach to improve outcomes in OAC is to increase early diagnosis as lower stage at diagnosis is associated with significantly improved 5-year survival [26]. While there is a lack of randomized controlled trials, observational studies have indicated that endoscopic surveillance in high risk BO patients is beneficial for an early diagnosis of OAC [27] although the cost-effectiveness of the endoscopic surveillance is debated [28] given the low annual conversion rate from BO to OAC [29]. To that end, the development of affordable, accurate diagnostic biomarkers could serve the following roles:(1)Early, accurate, and cost-effective diagnosis of BO and OAC(2)Reduce the need for endoscopy in the surveillance of individuals with low-risk BO(3)Identification of those with current HGD which would benefit from RFA(4)Enable broader population screening for those who do not meet the referral criteria for upper endoscopy.

In this section, we have assessed the emerging diagnostic biomarkers which could meet the clinical needs highlighted. The potential diagnostic biomarkers discussed are summarized in Table 2 below. Qin et al. performed whole-methylome sequencing on DNA extracted from frozen tissue samples of oesophageal cancer (OC) and controls to identify differentially methylated regions for use as biomarkers [30]. Of the regions identified, 12 demonstrating high discrimination between OC and control tissue were assessed in archived OC patient plasma (*n* = 76 OAC, *n* = 9 OSCC) and control plasma (*n* = 98). A panel of five differentially methylated sequences was found to accurately detect OC with a stage dependent sensitivity of 43% in stage I, 64% in stage II, 77% in stage III, and 92% in stage IV disease. Although site specificity was not investigated, it was anticipated that the differentially methylated regions may also be found in other GI tumors. The authors proposed that an OC-specific hypermethylated region such as DMRTA2 could be included to optimize specificity of assay. Other limitations of this study include OAC and OSCC being examined together, BO patient samples being excluded from the analysis (cross-reactivity with BO patients would limit the utility of the assay) and a largely uniform ethnic background of samples. However, the study did successfully demonstrate the feasibility and clinical utility of a panel of differentially methylated sequences as a means of non-invasive screening.

Since their discovery, there has been a great deal of interest in the role of non-coding RNAs (ncRNA) in disease pathology. Transcribed ultra-conserved regions (T-UCRs) are a subtype of ncRNAs which are composed of at least 481 genomic sequences more than 200 bp long and are absolutely conserved between mouse, rat and human genomes [31]. Fassan et al. used custom micro-RNA (miRNA) microarray chips to identify a panel of nine T-UCRs which were differentially expressed in NDBO (*n* = 14) when compared to normal oesophageal tissue (*n* = 14) [32]. Similarly, when the authors studied progression from normal oesophagus (*n* = 10) to NDBO (*n* = 10) to LGD (*n* = 10) to HGD (*n* = 10) to OAC (*n* = 10) in resected surgery specimens, they found 5 T-UCRs to be differentially expressed along the disease sequence. While these results are interesting, the T-UCRs identified would be of limited use as a diagnostic biomarker given the necessity to access tissue to measure expression. This lack of a translational approach also applies to other long non-coding RNAs (lncRNA) identified to be differentially expressed in normal/BO/OAC tissue [33,34]. However, it would be of interest to see if lncRNAs could play a role as a prognostic biomarker in BO or indeed, as a predictive biomarker for response to NA-T in OAC.

In recent years, several studies have examined the potential role of volatile organic compounds (VOC) for the diagnosis of OAC among other GI cancers. A 2016 pilot study investigated whether VOCs detected in the “headspace” of a plasma sample (the gaseous portion of the vial once equilibrium has been reached) could be used to accurately distinguish between GORD and OAC patients [35]. Samples were taken from 20 OAC patients prior to their staging endoscopy as well as 19 patients with GORD. The headspace of each participant’s plasma was analyzed for 22 pre-selected compounds in duplicate and nine VOC levels were found to be significantly different between the two groups (statistical significance was maintained after adjusting for age, sex and BMI). A diagnostic model to differentiate OAC from GORD was developed using just two VOCs (acrylonitrile and carbon disulfide) which provided a specificity, sensitivity, and positive and negative predictive values of 90%, 84%, 86%, and 89%, respectively. While this model would have mischaracterized 2/12 T3 tumors, it would have properly identified all 8 T1/2 stage tumors. This study provided a proof of feasibility that VOC headspace analysis could provide an affordable, non-invasive diagnostic tool for OAC screening. Limitations include the small sample sizes, the omission of a BO cohort and the fact that patients with atypical GORD symptoms were excluded despite the fact that more than 40% of those that develop OAC report no typical history of GORD symptoms [13].

Analysis of VOC profile in the breath has also been trialed as a potential diagnostic biomarker. Markar et al., in a multicenter study, aimed to validate a previously identified set of five VOCs measured from the breath, in the diagnosis of oesophagogastric cancer [36]. The breath samples from 162 non-metastatic oesophagogastric adenocarcinoma patients naïve to treatment and 163 patients with benign conditions or normal upper GIT was collected. Upon analysis, a sensitivity of 80% and a specificity of 81% for the diagnosis of oesophagogastric cancer was obtained. Inclusion of BO patients and validation of this method could aid screening in the primary care population presenting with upper GI symptoms. However, it should be noted that when compared to VOCs extracted from plasma, VOCs measured in the breath are likely to have a larger number of potentially confounding factors which could limit utility. These include the patient’s diet, microbiome and the environment the patient was in prior to the breath test [35].

The role of miRNA as biomarkers of disease has been increasingly recognized in recent years, with potential utility across a wide variety of pathologies, ranging from neurodegenerative disorders to cardiovascular disease [37]. In particular, exosomal miRNA detectable in serum have been coming under increased scrutiny in recent years for development of cancer biomarkers given their comparative resistance to RNase degradation. Chiam et al. analyzed serum samples from 47 individuals (19 healthy controls, 10 NDBO patients and 18 OAC patients with locally advanced or metastatic disease) focusing on the exosomal miRNA components [38]. Exosomal miRNA was profiled using the high throughput TaqMan OpenArray microRNA panel comprising 758 human miRNA probes. A panel of five miRNA ratios was found to distinguish between OAC and NDBO or healthy controls with an AUROC of 0.99. While promising, limitations of this study include an all-male cohort, no testing for cancer specificity and no early stage OAC or high grade BO samples included (which would benefit the most from early detection) [38]. Fassan et al. performed a similar investigation using serum from NDBO patients (*n* = 27), HGD patients (*n* = 8) and early OAC patients (T1a disease only; *n* = 10) [39]. Ten new miRNAs were found to be significantly (*p* < 0.01) differentially expressed in NDBO when compared with the other groups with an AUROC range of 0.75–0.83. Construction of a panel containing some permutation of these ten miRNAs could improve the diagnostic accuracy.

In a retrospective analysis, Campos et al. correlated neutrophil-leukocyte ratio (NLR) obtained from medical records of complete blood counts within 6 months of biopsy with diagnosis of NDBO (*n* = 72) to dysplastic BO (LGD or HGD, *n* = 11) to OAC (*n* = 30) [40]. NLR was found to increase significantly across the groups and, using an NLR cut-off of >2.27, was able to diagnose OAC with a sensitivity, specificity and AUROC of 80%, 71% and 0.8, respectively. Considering the wide-availability, low cost and non-invasive nature of NLR measurement, it represents an attractive biomarker. The retrospective nature, the low number of stage I/II patients included (*n* = 4) and the broad exclusion criteria represent important limits in this study.

Haboubi et al. accessed the use of the phosphatidylinositol glycan class A gene mutation assay, which measures erythrocyte mutations, for correlation with progression from GORD to BO to OAC [41]. The erythrocyte mutation frequency (EMF) of 192 patients with GORD (*n* = 77), NDBO (*n* = 62), LGD (*n* = 4), HGD (*n* = 11) and OAC (*n* = 42) and 137 healthy volunteers was accessed and correlated with histological status. Average EMF was significantly higher in HGD and OAC when compared to non-neoplastic cohorts (*p* < 0.001), suggesting that EMF could be measured to aid screening for high-risk disease. An increase in sample size and a longitudinal design would be required to validate the usefulness of this method.

Porter et al. identified high mobility group box-1 (HMGB1), a ubiquitous nuclear protein, as a novel potential diagnostic biomarker after analyzing protein expression in 218 human oesophageal tissue samples (including normal tissue *n* = 39, BO = 121 and OAC *n* = 58) via IHC [42]. Importantly, the tissue samples of BO included samples from NDBO, LGD, NDBO adjacent to dysplasia and NDBO adjacent to OAC. Target protein expression was assessed by two blinded independent observers using a semi-quantitative grading system (absent, weak, moderate or strong expression) in both cytoplasmic and nuclear compartments with conflicting scores resolved by discussion. Notably, there was an increased nuclear HMGB1 expression in the background BO of those with dysplasia or OAC when compared to BO from patients with non-dysplatic disease (71%, 67% and 27%, respectively). Similarly, there was a lower cytoplasmic HMGB1 expression in background BO of OAC tissue when compared to NDBO. Thus, IHC analysis for HMGB1 in BO biopsies could aid diagnostics as altered HMGB1 expression in BO could indicate missed dysplasia/OAC. This could be particularly useful in combination with novel technologies to non-invasively obtain oesophageal cell samples such as the Cytosponge [43].

**Table 2 cancers-14-03427-t002:** Summary table of BO and OAC/OC diagnostic biomarkers.

Biomarker Name	Disease State Tested	Biomarker Type	Regulation (in Disease State)	Sample Type	Testing Method	Reference
FER1L4, ZNF671, ST8SIA1, TBX15, ARHGEF4	OC	Diagnostic	↑ ↑ ↑ ↑ ↑	Plasma	Quantitative Methylation Specific PCR	[30]
Acrylonitrile, Carbon disulfide	GORD, OAC	Diagnostic	↓ ↓	Plasma	SIFT-MS	[35]
Butyric acid, Pentanoic acid, Hexanoic acid, Butanal, Decanal	Oesophagogastric Adenocarcinoma	Diagnostic	↓ ↓ ↑ ↑ ↑	Breath	SIFT-MS	[36]
RNU6-1/miR-16-5p, miR-25-3p/miR-320a, let-7e-5p/miR-15b-5p, miR-30a-5p/miR-324-5p, miR-17-5p/miR-194-5p	OAC	Diagnostic	↑ ↑ ↑ ↓ ↓	Serum	qPCR	[38]
miR-92a-3p, miR-151a-5p, miR-362-3p, miR-345-3p, miR-619-3p, miR-1260b, miR-1276, miR-381-3p, miR-502-3p, miR-3615	NDBO, HGD, OAC	Diagnostic	↑ ↑ ↑ ↑ ↑ ↑ ↑ ↓ ↓ ↓	Serum	qPCR	[39]
Neutrophil-leukocyte ratio	OAC	Diagnostic	↑	Serum	Whole Blood Count	[40]
Erythrocyte Mutation Frequency	HGD, OAC	Diagnostic	↑	Serum	Phosphatidylinositol glycan class A gene mutation assay (Flow cytometry)	[41]
HMGB1 (Nuclear, Cytoplasmic)	Dysplastic BO, OAC	Diagnostic	↑ ↓	Tissue Biopsy	IHC	[42]

Abbreviations: BO, Barrett’s Oesophagus; ELISA, Enzyme-linked Immunosorbent Assay; GORD, Gastro-oesophageal Reflux Disease; IHC, Immunohistochemistry; NDBO, Non-Dysplastic Barrett’s Oesophagus; OAC, Oesophageal Adenocarcinoma; PCR, Polymerase Chain Reaction; SIFT-MS, Selected Ion Flow Tube Mass Spectrometry.

## 4. Predictive Biomarkers of Neoadjuvant Treatment Response in OAC

While the treatment for early and advanced/unresectable OAC is well-defined (endoscopic resection and chemoradiotherapy, respectively), the optimum management of locally advanced disease is less clear [12]. Therapy for this cohort primarily focuses on neoadjuvant chemoradiotherapy (NA-CRT) or neoadjuvant chemotherapy (NA-C) followed by oesophagectomy [44]. However, with both neoadjuvant therapeutic modalities, pathological response only occurs in 25–30% of patients [45]. Thus, the majority of patients experience the toxicity associated with neoadjuvant therapy as well as a delay of surgery in exchange for little to no benefit, emphasizing the need for predictive biomarkers of response to NA-T.

P53 has already been discussed as a potential prognostic biomarker in BO, yet many have hypothesized that it could also play a role in predicting response to neoadjuvant treatment in OAC, partially due to the observed reduction in overall survival (OS) associated with p53 mutation as documented by a recent meta-analysis [46]. Kandioler et al. assessed p53 expression using a standardized gene-specific sequencing kit on DNA extracted from paraffin-embedded tissue of diagnostic tumor biopsies [47]. Patients went on to receive a neoadjuvant regimen of cisplatin and fluorouracil prior to surgical resection. This study included a cohort of 20 with OAC. Normal p53 expression was associated with improved tumor-associated survival in these patients. However, as association with tumor response to NA-T as per tumor regression grade (TRG) was not assessed, it is possible that normal p53 expression was associated with an improved response to surgery also.

In addition to its potential role as a diagnostic biomarker, NLR has also been investigated for its potential role as a predictive biomarker for response to chemotherapy in OAC [48]. Patients recruited were diagnosed with OAC (radiological TNM stage of I-III) which was deemed to be amenable to treatment with curative intent. Of 136 patients recruited, 23 (16.9%) were considered to have a good pathological response (TRG = 1/2 using the Mandard system) to NA chemotherapy (the majority of patients received cisplatin and 5-fluorouracil). Baseline NLR was strongly associated with pathological response to chemotherapy (TRG > 2) with an AUROC, sensitivity and specificity of 0.71, 80.5% and 60.9%, respectively, when a threshold for good vs. poor responders of NLR = 2.25 was chosen. Similarly, the 5-year OS rate for the low NLR group was 50% compared to 20% for high NLR. Limitations include the modest sample size and a selective cohort (only approximately one-quarter of patients diagnosed with OAC are deemed amenable to curative treatment). A noteworthy similar study by Noble et al. [49] did not find any association between NLR and TRG after epirubicin-based chemotherapy. However, increased preoperative NLR was associated with worse OS. Key differences between the studies include definition of good responders (the latter study included TRG 3) and the decision not to dichotomize the continuous NLR data on regression analysis.

Skinner et al. investigated the difference in miRNA expression of pre-treatment OAC tumor biopsies of NA-CRT (majority received 5-fluorouracil and a platinum compound) non-responders and responders as defined by presence or absence of tumor cells in resected specimen as determined by pathologist [50]. The study included three patient cohorts: discovery (*n* = 10), model (*n* = 43) and validation (*n* = 65). Pre-treatment biopsies of the discovery cohort (which included five responders and five non-responders) were evaluated for the expression of 754 different miRNAs of which 306 were found to be expressed in moderate to high levels. The 44 miRNAs most able to distinguish between responders and non-responders were assayed for in the model cohort. Four miRNAs (mir-505* (* refers to the alternative strand of miRNA), mir-99b, mir-451a, and mir-145*) were found to be significantly differentially expressed between responders and non-responders across both cohorts. Tumor samples with more of these miRNAs falling under the low-expression category were associated with a greater likelihood of being a responder in both the model and validation cohorts. By combining a score derived from expression of the four miRNAs of interest with clinical stage and tumor grade, a model was developed to predict the likelihood of a particular patient to respond to NA-CRT with an AUROC of 0.81. Strengths of this study include the validation of this model across different assay platforms and its three-cohort design. Limitations include the fact that only a subset of the miRNA profile of the tumor was assessed and the omission of survival outcomes.

Bibby et al. conducted a similar experiment by profiling the miRNA complement of 18 pre-treatment OAC biopsies and categorizing the cohort into responders and non-responders on the basis of post NA-CRT (cisplatin and 5-fluoruracil) TRG (responders TRG = one or two, non-responders TRG=4 or 5, TRG = three patients excluded) [51]. Of 742 miRNAs analyzed, 67 were found to be differentially expressed between responders and non-responders with Mir-330-5p being the most significantly altered between groups (downregulated in non-responders). Of note, Mir-330-5p was not included in the 306 of 754 miRNAs which demonstrated moderate to high expression in the study by Skinner et al. [50]. Potential reasons for this include the use of different miRNA profiling arrays and the different classification of responders vs. non-responders. Regardless, Bibby et al. concluded that Mir-330-5p could form part of a miRNA pre-treatment signature for stratifying patients based on likely response to NA-CRT.

In a comparable study, Chiam et al. used next-generation sequencing to analyze the miRNA expression profile of 31 pre-treatment OAC biopsy samples and correlated the results with the patient’s subsequent response to NA-CRT (as determined by AJCC tumor regression grading system) [52]. NA-CRT consisted of cisplatin and 5-florouracil chemotherapy with concurrent radiotherapy. A complete response (AJCC grade = 0, responders) was observed in nine while the other 22 patients had a partial or absence response (AJCC grade = 1–3, non-responders). Several predictive miRNA ratios were identified. In particular, miR-4521/miR-340-5p and miR-101-3p/miR-451a, which demonstrated a sensitivity and specificity of 95% and 89%, and 91% and 89%, respectively, also showed an association with relapse-free survival comparable to AJCC TRG. Given the sample size, verification of these results in independent cohorts is warranted.

Maher et al. used a SELDI-TOF-MS proteomic approach to identify novel predictive biomarkers of response to NA-CRT in OC. Immunoglobulin and albumin depleted pre-treatment serum from 16 responder (TRG 1 or 2) and 15 non-responder (TRG 3-5) OC patients were used in this study [53]. C3A and C4A levels were significantly higher in non-responders. Although this study included a limited number of OSCC in the sample, no significant differences were found when only OAC patients were accessed. Of note, Lynam-Lennon et al. identified mir-187 as one of 67 miRNAs differentially expressed between responders and non-responders (as per TRG) to cisplatin and 5-fluorouracil-based NA-CRT in pre-treatment OAC biopsy samples [54]. In vitro analysis demonstrated that overexpression of mir-187 resulted in down regulation of C3. Given that mir-187 was significantly lower in non-responders, the link between the two potential biomarkers is highlighted.

MacGregor et al. identified two more potential predictive biomarkers for OAC response to NA-CRT (oxaliplatin-fluorouracil based in this cohort), the proteins XPF and MUS81 [55]. These candidate biomarkers were identified following testing for association between the mRNA expression of 280 DNA repair genes in 38 pre-treatment biopsies and pathological response to chemotherapy (as determined by Mandard TRG), disease-free survival (DFS) and OS. Significant results were observed for several repair genes including ERCC1 (high levels associated with low DFS, lack of pathological response and poor OS) and EME1 (high levels associated with low DFS and lack of pathological response). As available antibodies used in IHC lack specificity for ERCC1 and EME1, their heterodimeric endonuclease partner proteins XPF and MUS81 were examined as surrogate markers. Low XPF levels were associated with pathological response to chemotherapy and high MUS81 levels were associated with worse 1-year DFS and OS. In order to ensure these results were predictive of response to chemotherapy and not simply prognostic of OAC outcomes, levels were examined in a retrospective surgery alone cohort (*n* = 54) with no association observed with OS. Further validation is required.

Borg et al. investigated the use of Podocalyxin-like protein (PODXL) as a predictive biomarker for (fluoro-pyrimidine- and oxaliplatin-based) NA-C response in OAC and gastric adenocarcinomas [56]. PODXL was selected for investigation as the group had previously linked expression to prognosis in a surgery-only cohort of OAC and GAC patients [57]. High PODXL expression in pre-treatment biopsy as determined by IHC was significantly associated with both lower post-neoadjuvant pathological T-stage and a superior histopathological response rate as determined using the Chirieac TRG system. Furthermore, using statistical comparison with the previously studied surgery only cohort, it was demonstrated that patients with PODXL-negative tumors received no obvious benefit from chemotherapy, in contrast to patients with PODXL-positive tumors. Study limitations include a heterogeneity of tumor sampling and the retrospective design.

Kukar et al. analyzed pre-treatment PET-CT standard uptake values for association with response to NA-CRT [58]. Unfortunately, the model produced was of limited utility as while it yielded a strong positive predictive value for incomplete pathologic response, the positive predictive value for complete pathological response was poor.

Lynam-Lennon et al. assessed the expression of four proteins (HSP60, ATP5B, GADPH and PKM2) linked to mitochondrial metabolism in 23 pre-treatment OAC biopsy samples for association with NA-CRT (cisplatin and 5-fluoruracil-based) response [59]. ATP5B expression (as measured by IHC and tissue microarray) was significantly higher in the tumors of poor responders (Mandard TRG = 3–5) when compared to good responders (TRG = 1–2), indicating potential utility as a predictive biomarker. However, expression was not found to be associated with survival data.

The potential predictive biomarkers discussed are summarized in Table 3 below.

## 5. Discussion

In this article, we have reviewed the recent research and collated the various biomarkers for BO/OAC in development, providing a broad overview of the ongoing work in the field. This collection of data provides a useful insight into the promising candidate biomarkers, the wide array of technologies used to identify them, and the difficulties faced in biomarker discovery and validation. By presenting an overview of biomarker development, we have also been able to identify common limitations of the existing research which can help to better design future studies. While the studies reviewed here represent good first steps for BO/OAC biomarker development, many obstacles remain between the current landscape and the validation of prognostic, diagnostic or predictive biomarkers for use in clinical practice.

Prognostic biomarkers promise to ease both the economic and patient burden of endoscopic surveillance in BO by allowing improved risk stratification, thereby enabling more targeted surveillance. Similarly, by identifying those at high risk of disease progression, RFA could be used to help reduce the number of patients who go on to develop OAC. However, the translation of prognostic biomarkers to clinic is hampered by the necessary longitudinal nature of studies and low rate of progressive disease [8]. As such, no biomarkers have replaced the gold standard of diagnosis of degree of dysplasia as a method of risk-stratification in BO. The current literature classifies non-progressors based on an arbitrarily decided length of follow-up, leading to inter-study heterogeneity and limiting comparability. While panels including a selection of biomarkers to predict risk of progression offer hope, these tend to be more technically difficult and expensive to implement. Finally, low statistical power due to small sample cohorts is a common limitation in the studies of prognostic biomarkers, further hindering progression to use in clinic.

Similarly, an obstacle in the development of diagnostic biomarkers for use in screening or to reduce the need for endoscopy in surveillance is the omission of a dedicated BO cohort in many of studies analyzed, thereby restricting the potential real-life utility of the identified biomarkers. Many studies conducted, particularly those investigating lncRNAs, also lacked a translational approach and although differential expression was identified along the BO-OAC disease sequence, potential utility as a diagnostic biomarker was lacking. Research on the relevance of lncRNAs as potential prognostic and predictive biomarkers in BO and OAC, respectively, would be of interest. Furthermore, the limited number of BO patients with HGD or early stage (T1) adenocarcinoma included in these studies represents another limitation for the development of diagnostic biomarkers for early detection of disease. Similarly, a number of studies failed to analyze LGD and HGD dysplasia as separate entities despite differing clinical management [3], therefore limiting the utility of results. In addition, a large portion of the appeal of a non-invasive diagnostic biomarker is for use in patient cohorts who would not typically qualify for endoscopic screening. This is particularly relevant, given that more than 40% of those who develop OAC do not experience typical symptoms associated with GORD and BO [13]. However, should a suitable non-invasive diagnostic biomarker be developed, the challenge of what cohorts of patients it should be used on remains, given the harm which can be associated with screening programs [60].

Several difficulties exist in comparing the existing literature regarding emerging predictive biomarkers for response to NA-T in OAC. First, the varying modalities of NA-T (including varying chemotherapeutic agent) limits their broad application to a specific NA-T regimen. Additionally, most analyzed studies used pathologic response as a surrogate for subsequent clinical outcomes. While pathological complete response (TRG1/2) has demonstrated an association with endpoints such as OS [61], a recent study found that among those patients considered pathologic non-responders, a substantial subset (21.3%) with lymph node downstaging (defined as regional lymph nodes which were disease positive on clinical staging having no evidence of disease during pathological staging) derived a survival benefit mirroring that of local response [62]. Thus, by associating potential biomarkers only with local pathologic response, many studies are omitting this cohort who seem to derive survival benefit from neoadjuvant treatment. Furthermore, while Mandard TRG was the predominant system used to grade pathologic response, numerous others were also employed, limiting the comparability of results. Similarly, the pathological response may be linked to time between NA-T and surgery [63]. However, many of the studies did not report time to surgery and among those that did, it varied from 2 to 8 weeks, therefore introducing a further potential confounding variable. A possible method to overcome this is by associating potential biomarkers with survival data in a second control cohort who do not undergo NA-T as in Borg et al. [56]. However, this method would be unethical in a non-retrospective design given the proven benefit of NA-T [12]. Development and use of a pathologic response grading system which accounts for lymph node downstaging alongside adjusting for time to surgery and incorporating survival data represents a valid strategy for standardizing results for predictive biomarker studies, thereby improving biomarker development.

## 6. Conclusions

Despite the wealth of research and the recent rise in incidence, patient outcomes in OAC have remained relatively poor [13]. Further independent studies in larger, prospective cohorts are needed to validate all biomarkers reviewed before they reach clinical practice. Of note, a prospective randomized trial in oesophageal cancer (with subgroup analysis by histological subtype) (NCT00525200), which will aim to determine if p53 mutation has a predictive role in response to three different types of NA-C, which will be randomly assigned, has recently completed recruitment. To our knowledge, this is the only ongoing clinical trial for a potential biomarker we have discussed. Regardless, the development of prognostic, diagnostic and predictive biomarkers promises to dramatically improve clinical management and outcomes for BO and OAC patients.

## Figures and Tables

**Figure 1 cancers-14-03427-f001:**
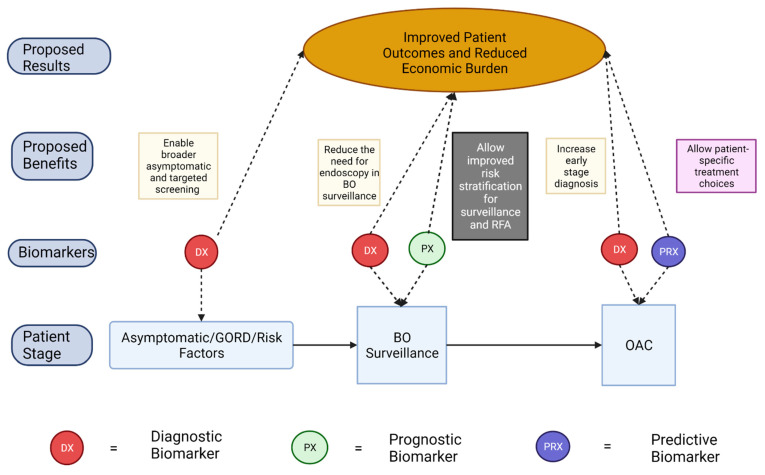
The influence of diagnostic, prognostic and predictive biomarkers on outcomes along the BO-OAC disease sequence.

**Table 1 cancers-14-03427-t001:** Summary table of prognostic biomarkers for risk of progression with BO.

Biomarker Name	Disease Progression Assessed	Biomarker Type	Regulation (in Progressors)	Sample Type	Testing Method	Reference
Aberrant P53 expression	NDBO/LGD-HGD/OAC	Prognostic	↑	FFPE Biopsy	IHC, FISH, NGS	[19,20,21]
Panel of abnormal P53 expression, abnormal AOL expression and diagnosis of LGD	NDBO/LGD-HGD/OAC	Prognostic	↑	FFPE Biopsy	IHC, pathologist review	[22]
MYC gain, p16 loss, aneusomy, age, circumferential BO length	NDBO-HGD/OAC	Prognostic	↑	Brush cytology specimens	FISH	[23]
8-oxo-dG	NDBO-HGD/OAC	Prognostic	↓	Tissue Biopsy	IHC	[25]

Abbreviations: AOL, Aspergillus Oryzae Lectin; FFPE, Formalin-Fixed Paraffin-Embedded Biopsy; FISH, Fluorescent In Situ Hybridization; HGD, High Grade Dysplasia; IHC, Immunohistochemistry; LGD, Low Grade Dysplasia; NDBO, Non-Dysplastic Barrett’s Oesophagus; NGS, Next-Generation Sequencing; OAC, Oesophageal Adenocarcinoma.

**Table 3 cancers-14-03427-t003:** Summary table of predictive biomarkers of response to neoadjuvant treatment in OC/OAC.

Biomarker Name	Disease	Therapy Type	Relative Regulation (Good Responders)	Sample Type	Testing Method	Reference
P53	OAC	NA-C	Not mutated	Tumor Biopsy	Gene Sequencing	[47]
Neutrophil-Leukocyte Ratio	OAC	NA-C	↓←→	Serum	Whole Blood Count	[48,49]
mir-505*, mir-99b, mir-451, mir-145*	OAC	NA-CRT	↓	Tumor Biopsy	qPCR	[50]
Mir-330-5p	OAC	NA-CRT	↑	Tumor Biopsy	qPCR	[51]
miR-4521/miR-340-5p, miR-101-3p/miR-451a	OAC	NA-CRT	↑	Tumor Biopsy	qPCR	[52]
C3A, C4A	OC	NA-CRT	↓	Serum (Ig and albumin depleted)	SELDI-TOF-MS	[53]
mir-187	OAC	NA-CRT	↑	Tumor Biopsy	RT-PCR	[54]
XPF, MUS81	OAC	NA-CRT	↓	Tumor Biopsy	IHC	[55]
Podocalyxin-like protein	OAC	NA-C	↑	Tumor Biopsy	IHC	[56]
ATP5B	OAC	NA-CRT	↓	Tumor Biopsy	IHC, RT-PCR	[59]

Abbreviations: Ig, Immunoglobulin; IHC, Immunohistochemistry; OAC, Oesophageal Adenocarcinoma; NA-C, Neoadjuvant Chemotherapy; NA-CRT, Neoadjuvant Chemoradiotherapy; PCR, Polymerase Chain Reaction; q, Quantitative; RT, Real Time; SELDI-TOF-MS, Surface Enhanced Laser Desorption/ionization Time-of-flight Mass Spectrometry; * refers to the alternative strand of miRNA.

## Data Availability

Not applicable.

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
