# Peer review of "Prognostic, Diagnostic and Predictive Biomarkers in the Barrett’s Oesophagus-Adenocarcinoma Disease Sequence"

_cancers, 2022, doi:10.3390/cancers14143427_

Round 1
Reviewer 1 Report
MAJOR COMMENTS
The study is a review that aims to summarise the recent studies looking at potential diagnostic and predictive biomarkers for barrett’s oesophagus and oesophageal adenocarcinoma. The authors have clearly demonstrated a clear understanding of the field and the current clinical needs and applications for the biomarkers. Key results for all the studies were presented. A summary of the current limitations was also discussed, which were reasonable and insightful. However, there are some key limitations and uncertainties in this review that needs to be addressed, particularly to ensure that that the review has accurately captured the key studies in the field.
11. I understand that this is meant to be a narrative review. However, it will be important to include some description on how the key studies were identified and included here. There are a lot of studies that seemed to be missing in this review. For instance, there are now emerging studies on the potential of lncRNAs but they were no included or discussed in this review. Many molecular biomarkers that were commonly raised in other reviews were also not captured e.g. DNA repairing genes including tp53 as a potential predictive biomarker for oesophageal adenocarcinoma. I am not suggesting that the review needs to be capture all studies in the field but there must be some reasons that certain key studies are excluded.
There are a lot of recent reviews published in similar topics hence I would caution the description that this is the first review undertaken (referring to line 88).
22. Distinction between “prognostic” biomarkers for Barrett’ oesophagus versus “diagnostic” biomarkers for oesophageal adenocarcinoma.
I admit this is a bit tricky and not as clear-cut. But is there a specific definition that distinguish the two? To me they kind of overlap. I understand that the prognostic biomarkers studies (Table 1) are mostly those that used BO and OAC samples while in table 2, the diagnostic studies may have included other samples such as normal controls or GORD. But the diagnostic biomarkers for oesophageal adenocarcinoma will need to have the capacity to distinguish between BO and OAC too.
I wonder If the type of disease progression assessed for each of the study cohort can be expanded in table 1 and 2 to help distinguish the studies in table 1 and Table 2. If not, they should be combined in a single table.
33. Including more details in tables.
One of the biggest obstacles in this area of research is the limited sample size in many biomarker studies. Thus, it will benefit for the readers to have this include in the table. Secondly, to have the general treatment information included in table 3. For instance, types of chemotherapy drugs and whether it is chemotherapy or chemoradiotherapy. This is quite important.
44. Figure 1. Strictly speaking, the review did not provide evidence on how the biomarkers can improve the outcomes. In the figure, I am assuming that the “effects” were the proposed outcomes of the biomarkers that will lead to the “improve patient outcomes and reduced economic burden”. Thus, the effects should really be presented just after “outcomes” and not “biomarkers”.
A few other questions have also arised from the figure. The diagnostic biomarkers were marked at all the disease states including the “asymptomatic/GORD/ risk factors”. Do you mean biomarkers that detects GORD? Secondly, it has been proposed that the diagnostic/prognostic biomarkers can be used to “replace endoscopy for surveillance” for BO patients. I think there needs to be caution with this statement. They may be used to triage patients for endoscopy but highly unlikely to replace it.
55. Dysplasia (ref 16) and LGD (ref 21) were stated as prognostic biomarkers identified for BO. This is inappropriate. Dysplasia and degree of dysplasia is known to be a histopathological indicator of likelihood to OAC. It is strictly speaking, not a biomarker.
66. Loss of p53 was mentioned to be associated with increased risk of progression from BO to OAC (ref 18) (refer to line 121). But this was stated as overexpression of p53 in table 1. Please clarify.
77. Please clarify the details described in the paragraph from line 205. Was the study (ref 28) based on tissue samples or plasma samples? line 206 suggested tissue but subsequent line 210 and in table 2 suggested plasma samples.
88. The studies looking at volatile compounds are quite exciting. It is more straightforward for the study looking at the VOC from patients breath but not so straightforward for the study looking at the VOCs of the headspace of plasma samples (ref 29). What does it actually mean? Isn’t it more tricky to develop a biomarker from VOC from plasma samples/ when you could have a more straightforward measurement of biomarkers using the plasma samples. Did the study propose why this is even feasible?
99. There seems to be little overlapping biomarkers identified, with the exception of a p53, NRL and miR-451 in the studies listed. P53 was mentioned to be currently used as a surveillance marker in the UK guidelines. Has this been tested in clinical trials or implemented in other international guidelines? In fact, are any of the biomarkers mentioned here for instance in clinical trials? For instance, the mirnas.
S Several mirnas were mentioned but there are very little overlap as mentioned. But some profiles included large groups of miRNAs identified in the studies. I wonder if we have missed out any overlapping miRNAs. Is there any advantage/ merit to compare overlapping miRNAs among these studies?
110. In general, biomarkers for BO and OAC are very much needed but the research in this area is hampered by so many difficulties. The discussion was well written and has highlighted the key limitations. However, despite efforts to develop biomarkers and studies reporting potential candidates, there seems to be little that went on to clinical trials and success. Though this is probably beyond the scope of the review, it may be especially insightful to note a few statements here if there are any near success or clinical trials that are awaiting outcomes in this space.
Reviewer 2 Report
Congratulations to the authors for the revision carried out.
All the sections of the article are written in a very detailed and understandable way. In the discussion, the discussion of predictive biomarkers of neoadjuvant treatment response in oesophageal adenocarcinoma is very well developed, but the discussion of the biomarkers of progression from Barret's oesophagus to oesophageal adenocarcinoma and the diagnostic biomarkers of Barret's oesophagus and oesophageal adenocarcinoma is not so clear.
Therefore, I believe that a new wording of the sections mentioned above is appropriate.
Author Response
Response to Reviewer 2 Comments
“Congratulations to the authors for the revision carried out.
All the sections of the article are written in a very detailed and understandable way. In the discussion, the discussion of predictive biomarkers of neoadjuvant treatment response in oesophageal adenocarcinoma is very well developed, but the discussion of the biomarkers of progression from Barret's oesophagus to oesophageal adenocarcinoma and the diagnostic biomarkers of Barret's oesophagus and oesophageal adenocarcinoma is not so clear.
Therefore, I believe that a new wording of the sections mentioned above is appropriate.”
Response: We thank the reviewer for their comment. The reviewers’ comments are appreciated and the discussion of prognostic and diagnostic biomarkers has been updated as suggested (lines 457-484) .
Reviewer 3 Report
Oesophageal adenocarcinoma (OAC) incidence has increased in the developed world. Barrett’s oesophagus (BO) is a precancerous condition of the oesophagus associated with chronic heartburn. This review focuses on potential biomarkers which could improve patient outcomes in BO/OAC. Given the absence of effective markers that improve patient outcomes along the BO-OAC disease sequence, the analysis and review of the few studies that explored the utility of serum and/or tissue biomarkers are especially interesting. This is a comprehensive and well-documented revision work that deserves to be published
Author Response
Response to Reviewer 3 Comments
“Oesophageal adenocarcinoma (OAC) incidence has increased in the developed world. Barrett’s oesophagus (BO) is a precancerous condition of the oesophagus associated with chronic heartburn. This review focuses on potential biomarkers which could improve patient outcomes in BO/OAC. Given the absence of effective markers that improve patient outcomes along the BO-OAC disease sequence, the analysis and review of the few studies that explored the utility of serum and/or tissue biomarkers are especially interesting. This is a comprehensive and well-documented revision work that deserves to be published.”
Response: Comments on the relevance and completeness of the work by reviewer 3 are appreciated.
Reviewer 4 Report
In this review authors discussed the emerging biomarkers that could improve patient outcomes along the BO-OAC disease sequence.
The manuscript is clear and generally well written. However, some points should be improved in order to further highlight the importance of this manuscript. In particular:
- In the paragraphs 3 and 4 authors did not report the emerging role of nicotinamide N-methyltransferase in esophageal squamous cell carcinoma (PMID: 34827592). Indeed, the enzyme was found to be upregulated in this neoplasm, making it a promising biomarker for diagnostic and prognostic purposes, as well as therapeutical target. Indeed, NNMT upregulation was found to be associated with esophageal squamous cell carcinoma cells to chemoresistance towards 5-fluorouracil (PMID: 32367570). Since several specific inhibitors have already been developed for this target (PMID: 34424711; PMID: 34572571; PMID: 34704059), the potential impact of this biomarker is evident. Authors should include this important topic through the text.
-Line 251-253: Authors should highlight the role of miRNA as biomarkers since miRNAs showed a diagnostic predictivity also in non-cancerous disease (PMID: 32726711, 35615626 and 35563107)
Author Response
Response to Reviewer 4 Comments
In this review authors discussed the emerging biomarkers that could improve patient outcomes along the BO-OAC disease sequence.
The manuscript is clear and generally well written. However, some points should be improved in order to further highlight the importance of this manuscript. In particular:
- In the paragraphs 3 and 4 authors did not report the emerging role of nicotinamide N-methyltransferase in esophageal squamous cell carcinoma (PMID: 34827592). Indeed, the enzyme was found to be upregulated in this neoplasm, making it a promising biomarker for diagnostic and prognostic purposes, as well as therapeutical target. Indeed, NNMT upregulation was found to be associated with esophageal squamous cell carcinoma cells to chemoresistance towards 5-fluorouracil (PMID: 32367570). Since several specific inhibitors have already been developed for this target (PMID: 34424711; PMID: 34572571; PMID: 34704059), the potential impact of this biomarker is evident. Authors should include this important topic through the text.
Response: While we thank the reviewer for their comments, we believe that including the emerging role of nicotinamide N-methyltransferase in esophageal squamous cell carcinoma would be beyond the scope of this review, given that the focus is on biomarkers in the Barrett`s oesophagus-adenocarcinoma disease sequence, not including esophageal squamous cell carcinoma.
-Line 251-253: Authors should highlight the role of miRNA as biomarkers since miRNAs showed a diagnostic predictivity also in non-cancerous disease (PMID: 32726711, 35615626 and 35563107)
Response: We thank the reviewer for their comment. A sentence about the role of miRNA as biomarkers has been added highlighting the expanding role of miRNA as biomarkers (line 271-273).